# The Bowel-Associated Arthritis–Dermatosis Syndrome (BADAS): A Systematic Review

**DOI:** 10.3390/metabo13070790

**Published:** 2023-06-25

**Authors:** Italo Francesco Aromolo, Domenico Simeoli, Carlo Alberto Maronese, Andrea Altomare, Daniele Noviello, Flavio Caprioli, Angelo Valerio Marzano

**Affiliations:** 1Dermatology Unit, Scientific Institute for Research, Hospitalization and Healthcare Ca’ Granda Ospedale Maggiore Policlinico, 20122 Milan, Italy; italo.aromolo@unimi.it (I.F.A.); domenico.simeoli@unimi.it (D.S.); carlo.maronese@unimi.it (C.A.M.); 2Department of Pathophysiology and Transplantation, University of Milan, 20122 Milan, Italy; daniele.noviello@unimi.it (D.N.); flavio.caprioli@unimi.it (F.C.); 3Scientific Institute for Research, Hospitalization and Healthcare Istituto Ortopedico Galeazzi, 20122 Milan, Italy; andrea.altomare@grupposandonato.it; 4Gastroenterology and Endoscopy Unit, Fondazione Scientific Institute for Research, Hospitalization and Healthcare Ca’ Granda Ospedale Maggiore Policlinico, 20122 Milan, Italy

**Keywords:** bowel, dermatosis, arthritis, neutrophilic dermatoses, inflammatory bowel diseases

## Abstract

Bowel-associated arthritis–dermatosis syndrome (BADAS) is a rare neutrophilic dermatosis that was first described in 1971 in patients who underwent bypass surgery for obesity. Over the years, the number of reported cases associated with medical gastroenterological conditions, particularly inflammatory bowel disease (IBD), has progressively increased. To date, there are no systematic reviews in the literature on BADAS. The design of an a priori protocol was based on PRISMA guidelines, and a search of PubMed and Scopus databases was conducted for articles published between 1971 and 2023 related to the topic. Fifty-one articles including 113 patients with BADAS were analyzed in this systematic review. Bariatric surgery and IBD were the most frequently reported causes of BADAS, accounting for 63.7% and 24.7% of all cases, respectively. A total of 85% of cases displayed the typical dermatological presentation, including urticarial maculopapular lesions centered by a vesicopustule, with the majority of lesions located on the upper limbs (73.5%). Polyarthralgia or localized arthritis were always present. Atypical presentations included cellulitis-like, erythema-nodosum-like, Sweet-syndrome-like and pyoderma-gangrenosum-like manifestations. Gastrointestinal symptoms were frequently observed in IBD-related cases (67.9%). The histopathology showed a neutrophilic infiltrate (96.6%). The most commonly used treatment regimens consisted of systemic corticosteroids, metronidazole and tetracyclines, either alone or in combination. A relapsing–remitting course was observed in 52.1% of patients. In conclusion, BADAS is a neutrophilic dermatosis that presents with a wide variety of cutaneous manifestations, both typical and atypical. Gastrointestinal symptoms are frequently observed, particularly in cases related to IBD. The histopathology is clear but not specific compared with other neutrophilic dermatoses. The diagnosis can be challenging, but the relapsing–remitting course and the strong association with polyarthralgia and gastrointestinal disease can aid in the diagnosis.

## 1. Introduction

Bowel-associated dermatosis–arthritis syndrome (BADAS) is an uncommon neutrophilic dermatosis presenting with recurring cutaneous and articular manifestations in patients with gastrointestinal disease. Histologically, it presents with a neutrophilic infiltrate indistinguishable from that of other neutrophilic dermatoses. It was originally described in 1971 by Shagrin et al. in patients who had undergone bowel bypass surgery for obesity, taking the name of “bowel bypass syndrome” [1]. Over the years, the number of published cases associated with medical rather than surgical gastroenterological conditions, particularly inflammatory bowel disease (IBD), has progressively increased. 

There are no systematic reviews in the literature about BADAS. The aim of this article is to summarize and critically evaluate the existing literature on BADAS, providing a comprehensive overview of the current understanding of this disorder, including its epidemiology, clinical manifestations, histopathology and management strategies.

## 2. Materials and Methods

### 2.1. Protocol and Literature Search

The recommendations contained in the preferred reporting items for systematic reviews and meta-analyses (PRISMA) statement were followed. The review protocol was registered on PROSPERO (CRD42023426082). The review was carried out by searching both the PubMed and Scopus databases using the search terms “bowel AND dermatosis AND arthritis AND syndrome” and “BADAS” and “Bowel AND Bypass AND Syndrome”. The search included publications from 1 January 1971 to 28 February 2023 and was independently performed and verified by two researchers (IFA and DS).

### 2.2. Selection of Articles

Articles were evaluated based on their title and abstract, and those deemed relevant were further assessed in full text. If there was a disagreement about the relevance of an article, it was resolved by a third, independent author (AVM). The articles were included in the data extraction and analysis if the diagnosis was based on the following criteria: (a) a compatible cutaneous presentation, (b) localized arthritis/polyarthralgia, (c) a history of gastrointestinal disease, and (d) the absence of an alternative diagnosis. 

### 2.3. Data Extraction and Analysis

Four authors (IFA, DS, DN and CAM) critically reviewed the included articles and independently extracted the following variables onto a Microsoft Excel spreadsheet: sex, age, localization, appearance and course of skin lesions, medical or surgical gastrointestinal comorbidities, latency from onset of gastrointestinal disease/surgery to cutaneous manifestations, cutaneous symptoms, presence of localized arthritis, sites of frank arthritis, presence of polyarthralgia, fever, gastrointestinal symptoms, histopathology of cutaneous lesions, direct immunofluorescence, blood tests, treatments and response to treatments. Any disagreement in the extraction of the variables was resolved through discussion with an independent author (AVM).

Categorical variables were reported as frequencies and percentages, whereas continuous variables were reported as means and standard deviations (SDs). Relevant data were not available for every patient; percentages refer to the number of patients for whom information about a specific parameter was available or inferable.

The data were synthesized using Excel functions and presented in the attached tables.

### 2.4. Quality, Limitation and Risk of Bias

Two authors (IFA and DS) assessed the methodological quality of the evidence and risk of bias of the included studies independently, according to previously published criteria for case reports and series [2]. Any disagreement was resolved through discussion with a third author (AVM). The under-recognition and under-publication of BADAS may impact the reliability of the presented data (detection and publication biases). Moreover, the reported cases often had imprecise descriptions and were not always complete relative to the investigated variables: in these cases, patients were considered in the qualitative synthesis only limited to available data (e.g., histopathology was performed in just 51% of patients). In some case series, parts of the data were excluded due to their qualitative nature (e.g., Stein et al. [3] reported data from a population that included both patients with bowel-associated arthritis–dermatosis and those with joint involvement without cutaneous manifestations, without distinguish between them). The heterogeneity in data presentation across different studies was a limitation during data extraction (e.g., not all articles clearly used terms such as complete remission, partial remission or active disease in assessing the treatment response). Finally, the overlap of features with other neutrophilic dermatoses may have resulted in misdiagnosis, leading to an overestimation or underestimation of the number of cases.

## 3. Results

### 3.1. Identification of Eligible Articles

As shown in the PRISMA flow diagram (Figure 1), the literature search identified references (PubMed = 381; Scopus = 233). Additional records (*n* = 2) were identified through other sources (e.g., references). After removing duplicates (*n* = 232) and articles deemed not relevant through title and abstract screening (*n* = 328), 56 articles were evaluated in full text. Among them, 51 articles (comprising 42 individual case reports and 9 case series) fulfilled the eligibility criteria [3,4,5,6,7,8,9,10,11,12,13,14,15,16,17,18,19,20,21,22,23,24,25,26,27,28,29,30,31,32,33,34,35,36,37,38,39,40,41,42,43,44,45,46,47,48,49,50,51,52,53] and were incorporated into the synthesis (Appendix A), resulting in a final sample of 113 patients being analyzed.

### 3.2. Demographic and Clinical Features

The review includes 113 patients. The majority of them are female (79.2%), with a mean age of 39.7 years. Bariatric surgery and IBD are the most frequently associated diseases, accounting for 63.7% and 24.7% of all cases, respectively. The typical dermatological presentation consists of urticarial maculopapular lesions centered by a vesicopustule (85% of cases), with the majority of lesions affecting the upper limbs (73.5%). Polyarthralgia or localized arthritis were always present. Atypical presentations include erythema-nodosum-like (16 patients), urticarial Sweet-syndrome-like (6 patients), cellulitis-like (2 patients) and pyoderma-gangrenosum-like (2 patients) manifestations. Gastrointestinal symptoms are frequently observed in IBD-related cases (67.9%). The course is often relapsing–remitting; a history of previous episodes is reported by 52.1% of patients. Demographic and clinical data are summarized in Table 1 and Table 2. 

### 3.3. Histopathological, Immunopathological and Laboratory Tests

Skin histopathology is not specific compared with other neutrophilic dermatoses; the most commonly reported findings are neutrophilic dermal and/or epidermal infiltrate (96.6%) and papillary dermis edema (34.5%). Direct immunofluorescence is positive in 65.7% of the investigated patients; C3 and IgG are the most frequently reported deposits (48.6% and 34.3%, respectively). Histopathological and immunopathological features, as well as other laboratory data, are summarized in Table 3. 

### 3.4. Treatment and Clinical Outcome

Systemic corticosteroids, metronidazole and tetracyclines, either individually or in combination, are the most commonly used drugs for the management of a single episode. However, long-term remission often requires treatment of the underlying disease: pharmacological treatments for IBD-related cases (sulfasalazine, azathioprine, mycophenolate mofetil, cyclosporin A, infliximab, ustekinumab, ruxolitinib) and surgical revision for cases related to prior surgery. A total of 84.5% of patients achieve complete remission (CR). Global treatment modalities and clinical outcomes are summarized in Figure 2. Treatment modalities and clinical outcomes grouped by the associated condition are summarized in Appendix A (IBD associated) and Appendix A (post-surgical).

## 4. Discussion

BADAS is a rare disease that presents with recurrent, sterile, vesicopustular lesions and articular pain, mostly in female patients with gastrointestinal conditions. From a dermatological perspective, individual manifestations may be classified within the spectrum of neutrophilic dermatoses, consisting mainly of Sweet syndrome, pyoderma gangrenosum and neutrophilic dermatosis of the dorsal hands. Historically, the disease was linked to surgery, particularly bariatric surgery, but has since been connected to a variety of medical gastrointestinal conditions, especially IBDs including Crohn’s disease and ulcerative colitis (UC). The worldwide prevalence of IBD has nearly doubled in the period from 1990 to 2017, increasing from 3.7 million to 6.8 million affected individuals. As a result, the epidemiology of BADAS has also changed [54].

### 4.1. Epidemiology, Etiology and Associated Diseases 

BADAS occurs most commonly in female individuals (79.8%) in the fourth decade of life. Pediatric cases are rare (5.2%). BADAS is more commonly associated with UC (57.3% of cases) and should be proactively investigated as other extraintestinal manifestations [55]. Bariatric surgery and IBD are by far the most commonly causes of BADAS, accounting for 63.7% and 24.7% of all cases, respectively. Rarely, other medical or surgical gastrointestinal conditions have been reported to be associated with the disease, such as appendicitis, diverticulitis, cystic fibrosis, surgery for gastroduodenal peptic ulcers, trauma and congenital aganglionic megacolon. With modern surgical procedures, the incidence of surgery-related BADAS has dramatically decreased. Only 9 out of 81 cases of surgery-related BADAS have been published after 1990. Surgical procedures associated with BADAS include jejunoileal bypass [5], jejunocolic bypass [5], ileocolic bypass [5], Billroth II gastrectomy [10], Roux-en-Y jejunectomy [17], ileoanal pouch anastomosis [23], biliopancreatic diversion [26] and laparoscopic gastric bypass [37].

### 4.2. Pathogenesis

Currently, the pathogenesis of BADAS is not fully understood. It was comprehensively reviewed by Carubbi et al. in 2013 [56]. In the context of gastrointestinal disease, overgrowth and subsequent translocation of gut bacteria can cause an abnormal activation of the local immune response, later leading to systemic disease. The first event in the pathophysiological scenario of BADAS consists of the production of immune complexes in the context of gut inflammation. These access the bloodstream and deposit in tissues such as the skin and joints, leading to the activation of both classic and alternative complement pathways. Fragments of complement molecules, such as the anaphylatoxin C5a, promote neutrophil migration, activation and exocytosis in tissues. Growth factors (e.g., granulocyte colony-stimulating factor (G-CSF)), adhesion molecules (selectins, integrins), interleukin (IL)-8 and tumor necrosis factor alpha (TNF-α) appear to be involved in promoting this neutrophil-rich inflammation [29]. Although the deposition of immune complexes is primarily vascular, upon histopathology diffuse neutrophilic infiltrates are recognizable in the dermis and findings of true vasculitis (e.g., fibrinoid necrosis) are present in less than a quarter of cases.

Evidence that supports the role of immune complexes in the pathogenesis of BADAS includes: (i) the fact that circulating immune complexes have been documented in the sera of patients [7,57,58]; (ii) the detection of serum cryoglobulins in a proportion of patients [7,59]; (iii) the frequent demonstration of deposits of Ig and C3 in skin biopsies on direct immunofluorescence (see Table 3); and (iv) the fact that perivascular deposits of Ig and C3 have been found in the synovial tissue of a patient with BADAS on direct immunofluorescence [57].

An activation of both the classic and alternative pathways of the complement system has been demonstrated in BADAS by Moller et al. [60]. Indeed, serum levels of complement components as well as their ratios (C3:C4, C3:C3PA, C3PA:C4) were abnormal in 6 out of 17 patients. Complement proteins opsonize bacteria, facilitating their phagocytosis through complement receptors, and promote the local inflammation by acting as chemotactic factors [61].

Undoubtedly, gut bacteria and their components are other important players in the pathogenesis of BADAS. In fact, peptidoglycan—which is a polysaccharide that forms a rigid envelope around the bacterial cytoplasmic membrane—seems to be the most important antigen within immune complexes.

Skin testing with *Streptococcus pyogenes* antigens causes an exacerbation of BADAS and may even provoke the onset of the syndrome in patients who underwent ileojejunal bypass surgery [62]. Animal experiments demonstrated that injecting whole bacteria can lead to a BADAS-like phenotype [63,64,65]; similar manifestations were seen when purified peptidoglycan was injected in animal models [66,67]. Other laboratory studies conducted in humans support the role of intestinal bacteria. Indeed, circulating complexes consisting of IgG immunoglobulins directed against *Escherichia coli* and *Bacteroides fragilis* have been detected [59]. On direct immunofluorescence, an anti-E. coli antiserum has been shown to stain the dermoepidermal junction in skin biopsies of BADAS patients with granular IgG deposits [7].

More recently, the role of players other than immune complexes has emerged in the pathogenesis of BADAS. Gut bacterial pathogen-associated molecular patterns (PAMPs) activate intestinal epithelial cells and the local innate immune response through Toll-like receptors and NOD-like receptors, leading to the overproduction of cytokines such as IL-1 and IL-18 [56]. This overproduction may act both locally and systemically, likely contributing to other BADAS manifestations too. It is worth underscoring that IL-1 has a central role in the pathogenesis of neutrophilic dermatoses [68]. Gut bacterial PAMPs appear to contribute to the pathogenesis of BADAS through molecular mimicry with self-antigens, inducing both T-cell-mediated and B-cell-mediated autoimmunity [56]. Vitamin D deficiency is another factor proposed to contribute to BADAS pathogenesis. Vitamin D deficiency due to malabsorption may frequently be found in patients affected by the gastrointestinal conditions associated with BADAS. The immunomodulatory role of the active form of vitamin D is well recognized and its deficiency has been associated with the risk of developing autoimmune diseases. Vitamin D receptor is expressed on many immune cells and, among its functions, downregulates the activity of T-helper 1 and T-helper 17 lymphocytes [69]. Moreover, vitamin D protects the gut barrier by regulating tight junction proteins and inhibiting intestinal apoptosis [70].

Gut permeability alteration and dysbiosis play a role in the pathogenesis of BADAS, especially in IBD-related cases. The typical dysbiosis observed in IBD is characterized by an increase in the populations of Proteobacteria and Fusobacteria, along with a reduction of Firmicutes [71]. An increase in the Bacterioides group has also been demonstrated in IBD [72]. Indeed, circulating antibodies against B. Fragilis have been detected in patients with BADAS [59].

The alteration of intestinal permeability is an early event in the pathogenesis of inflammation in IBD. Pathological changes in cellular tight junctions and mucus layer cause microorganisms to penetrate the epithelial barrier and trigger an exaggerated immune response. In mice lacking proper glycans, the protective mucus layer is depleted, resulting in increased permeability of the intestinal mucosa and a susceptibility to IBD [73]. Similarly, mice models knocked out for the MUC2 gene (which codes for Mucin2, another component of the mucosal barrier) spontaneously develop colitis [74]. 

Exosome-like nanoparticles may also a have a role in controlling the intestinal barrier. These small nanovesicles, enclosed by a lipid bilayer membrane and containing cytosolic components such as messenger RNA and microRNA, are found in plants and food. In a mice model, it has been demonstrated that exosome-like nanoparticles derived from ginger activate the aryl hydrocarbon receptor (AhR)-mediated pathway, which, in turn, increases IL-22 production [75]. IL-22 strengthens the intestinal barrier and has been shown to alleviate colitis in mice models [76].

IL-10 is another metabolite in IBD inflammation. It stimulates the production of intestinal mucus, downregulates the expression of major histocompatibility complex II and suppresses other inflammatory cytokines [77,78,79]. Mice deficient in IL-10 develop spontaneous colitis if they are not kept in a germ-free environment, and this colitis improves when IL-10 is administered [80]. 

Tryptophan, short-chain fatty acids (SCFAs) and bile acids are additional metabolites involved in gut homeostasis and their levels are altered in intestinal diseases. 

Tryptophan plays a role in intestinal inflammation and the maintenance of the epithelial barrier. Higher levels of tryptophan have been associated with lower IBD activity [81], and an ongoing study is testing its oral administration in affected patients [82]. Tryptophan is metabolized into indole derivatives by specific gut bacteria; these metabolites activate AhR, increasing levels of IL-22 and strengthening intestinal barrier function. Tryptophan levels are lower in patients with IBD due to changes in gut bacteria capable of metabolizing it [83]. 

SCFAs, primarily produced in the cecum and proximal colon by certain bacteria, have anti-inflammatory properties and are considered a promising supplementary treatment in IBD. They bind to SCFA receptors and cause a downstream activation of the inflammasome–IL-18 axis [84]. Dysbiosis in IBD patients is associated with a decrease in the number of SCFA-producing bacteria [85]. Bile acids are another class of anti-inflammatory metabolites that regulate gut homeostasis. In IBD, they are reduced due to malabsorption caused by ileal involvement. Moreover, there is a significant reduction in secondary bile acids, which correlates with alterations in the microbiome and a reduced bacterial deconjugation activity [86,87]. Secondary bile acids bind to the bile receptor TGR5, leading to a reduction in IL-6 and TNF α and an increase in IL-10 [88].

The role of diet has not been investigated in BADAS. However, some considerations can be made on studies conducted on IBD, to which BADAS is closely related. In particular, numerous studies have shown that the low FODMAP (fermentable oligosaccharides, disaccharides, monosaccharides and polyols) diet (LFD) is capable of reducing symptoms in patients with IBD, including abdominal pain, bloating and flatulence [89,90,91]. FODMAP carbohydrates are believed to be poorly absorbed by the small intestine, so their fermentation exacerbates gastrointestinal symptoms in diseases such as IBD and, possibly, BADAS [92]. However, a pathogenetic LFD role in regulating IBD activity has not been demonstrated.

### 4.3. Clinical Manifestations

In this systematic review of 113 BADAS cases, the cutaneous picture was typified by urticarial maculopapules surmounted by central vesicopustules that later evolved into erosions and then into crusts, healing without scars. Importantly, the lesions were always sterile. Upper limb involvement was predominant (73.5%), whereas the lower limbs and the trunk were less frequently affected. The face was only rarely involved (18.1%), with anecdotal lip lesions (three patients). Cutaneous lesions were always accompanied by localized arthritis or polyarthralgia. When present, localized arthritis more commonly affected the upper extremities. The majority of patients did not experience cutaneous symptoms such as pain or itching, although the anamnestic differentiation between skin and joint pain was not always reliable. Expectedly, the presence of gastrointestinal symptoms was more common in IBD-associated cases (67.9%) than non-IBD ones (23.5%). Moreover, BADAS appeared to parallel IBD activity, similarly to other neutrophilic dermatoses such as Sweet syndrome, but differently from other entities of this group, such as pyoderma gangrenosum [93]. When a patient with IBD shows new skin lesions, it is important to investigate any gastrointestinal manifestation such as diarrhea, pain, hematochezia or melena. Systemic involvement in BADAS is frequent but not constant: fever is present in about half of the cases, leukocytosis and/or neutrophilia in 40.3% and elevated erythrocyte sedimentation rate (VES) in 66.6%.

The disease often has a relapsing–remitting course that extends over several months or even years, with each episode lasting a few days. Patton et al. clearly described the evolution of cutaneous lesions [94]. These begin as 3 to 10 mm erythematous macules that develop into papular, vesicular and then pustular 2 to 4 mm lesions over 1 or 2 days. Overall, lesions last 2 to 8 days, recurring approximately every 1 to 6 weeks [33].

A thorough gastrointestinal history should be obtained from patients presenting with this cutaneous–articular picture: in about 25% of cases the onset of the cutaneous–articular component of BADAS follows that of gastrointestinal disease or surgery by over 5 years. However, the absence of gastrointestinal disease at the time of first evaluation should not rule out the diagnosis of BADAS completely. Indeed, the latter represents the first presentation of IBD in 33% of IBD-related cases. 

Atypical presentations of BADAS are possible. In some patients, purpuric features have been described in otherwise typical lesions. Cellulitis-like lesions on the lower limbs (two patients [28,30]) as well as erythema-nodosum-like lesions (16 patients) have been reported, either alone or in association with typical lesions. These should raise the suspicion of BADAS in patients with gastrointestinal diseases and skin lesions, regardless of their morphology. Two patients [37,52] presented with painful ulcerated nodules and plaques on the dorsal aspect of the hands in the absence of any other lesion, which is a picture strongly reminiscent of neutrophilic dermatosis of the dorsal hands. Six patients showed persistent urticaria-like lesions without vesicopustular lesions, in keeping with the classic presentation of Sweet syndrome. Two patients presented pyoderma-gangrenosum-like lesions in addition to the typical vesicopustules [16,23]. These observations further underscore the concept that different neutrophilic forms may coexist in the same patient, strengthening their inclusion in the same spectrum.

### 4.4. Histopathology

The histopathological features of BADAS are not specific respective to other neutrophilic dermatoses. The microscopic picture dynamically reflects the clinical appearance of biopsied lesions [6,62]. Early macules show vasodilation, perivascular edema and a mild neutrophilic infiltrate in the dermis, with the epidermis still appearing normal. In the papular stage, dermal edema and neutrophilic infiltrates become more abundant and diffuse. In pustular lesions, the neutrophilic infiltrate extends into the epidermis, forming a subcorneal pustule. In this stage, the epidermis may show necrotic keratinocytes and intercellular edema. In older papulopustules, neutrophils fragment into debris and lymphocytes, macrophages and eosinophils may appear (9% of cases according to our review). Expectedly, erythema-nodosum-like lesions present septal inflammation. Additional features such as signs of vasculitis (22.45%) and neutrophil adnexotropism (5%) are also possible. However, the inflammatory density does not make it easy to identify true adnexotropism. Furthermore, an atypical histopathological picture has been reported, consisting of a subcorneal pustule formation without significant dermal neutrophilic infiltrates [33].

Direct immunofluorescence on skin biopsy is often positive (65.7%) but non-specific. The C3 complement fraction is the most commonly detected antigen (48.6%), both along the basement membrane zone and on the walls of dermal vessels. IgG and IgM deposits are also frequently reported (34.3% each), this being consistent with the abovementioned role of immune complexes in the pathogenesis of BADAS.

### 4.5. Differential Diagnosis

The differential diagnosis from other neutrophilic dermatoses (e.g., neutrophilic dermatosis of the dorsal hands or Sweet syndrome) is not always clear. The strong association with an underlying gastrointestinal disorder or surgery, the presence of polyarthralgia and the typical relapsing–remitting course may play a decisive role in correctly diagnosing BADAS, especially when it comes to differentiating it from other entities in the presence of an identical clinical–pathological picture on the skin. In IBD-related cases, the differential diagnosis should always include other cutaneous manifestations of IBD, such as pyoderma gangrenosum, erythema nodosum or Sweet’s syndrome [93]. Despite the classic evolution into ulcers with an undermined and violaceous border, early pyoderma gangrenosum lesions may represent a challenge in the differential diagnosis, especially when considering its pustular variant. The presence of fever and pustular lesions mandates exclusion of an infectious cause, such as gonococcal septicemia, systemic candidiasis or subacute endocarditis. Cutaneous and blood cultures are negative in BADAS. Both skin-limited and systemic IgA vasculitis can also be considered among differential diagnoses, especially when purpuric features are noticeable. Although concurrent involvement of the skin, joints and gastrointestinal system may be documented in both, a predominant histological picture of leukocytoclastic vasculitis in postcapillary venules is seen in IgA vasculitis, with classic IgA deposition on immunofluorescence.

Urticarial maculopapules centered by an erosion in exposed body sites may be reminiscent of arthropod bites; however, the presence of systemic involvement and the absence of pruritus are usually sufficient to rule it out. Interestingly, in children BADAS is frequently misdiagnosed as varicella-zoster virus infection or coxsackie virus exanthems.

### 4.6. Treatment

In many cases, single episodes resolve spontaneously over the course of some days [94]. Conversely, active treatment of BADAS is necessary in patients with severe and/or persistent disease. Currently, the therapeutic approach is empirical and based mainly on systemic corticosteroids and antibiotics, either alone or in combination. Antibiotics reduce the gastrointestinal bacterial load as well as bacterial translocation, whereas systemic corticosteroids downregulate the immune response both at a local and systemic levels. The most commonly used antibiotics are tetracyclines, such as doxycycline, and metronidazole. Although tetracyclines have well-known anti-inflammatory properties [95], metronidazole is highly active against Gram-negative anaerobic bacteria of the gut, such as *B. fragilis*, whose important role in the pathogenesis of BADAS has already been discussed [96]. Dapsone is also frequently used for its anti-neutrophilic action in addition to its antimicrobial one. Cyclic administration of different antibiotics, e.g., for one week each month, has been considered given the relapsing–remitting course of the condition [56]. Non-steroidal anti-inflammatory drugs may help control arthralgias [7]. Finally, topical corticosteroids are rarely used alone due to their lack of effectiveness on the systemic component of the disease.

Remission of an episode after treatment with antibiotics and/or corticosteroids is often temporary. Treatment of the underlying disease is essential for achieving complete remission in the long term, including pharmacological treatment for IBD-related cases and surgical restoration of normal bowel anatomy for surgery-related cases. Treatment of the underlying IBD in the context of BADAS includes a wide variety of drugs such as sulfasalazine, steroid sparing immunosuppressants (azathioprine, mycophenolate mofetil, cyclosporin A), biological agents (infliximab, ustekinumab) and small molecules (ruxolitinib). It is impossible to dissect the efficacy of these drugs on the underlying IBD from their efficacy on BADAS. However, some of these drugs, such as the anti-TNF-α agents (e.g., infliximab) and the IL-12/23 inhibitor ustekinumab, have been used successfully in other neutrophilic dermatoses, such as pyoderma gangrenosum [97]. Likewise, their use in BADAS could be hypothesized, paving the way for new therapeutic possibilities also in non-IBD-related cases.

Overall, the response to therapy is quite good, with 84.5% of patients (including both surgical and IBD-related cases) reaching CR. Unfortunately, data on follow-up are scant and mostly short term. Recurrences are reported in only 4.9% of patients who achieved CR, although maintenance pharmacotherapy is often necessary.

## 5. Conclusions

In conclusion, we provided an overview of the literature on BADAS, defining epidemiology, clinical manifestations, histopathology and treatments as comprehensively as possible. BADAS is a neutrophilic dermatosis presenting with a wide variety of cutaneous manifestations, ranging from the typical urticarial papules centered by a pustule to atypical presentations such as cellulitis-like or erythema-nodosum-like lesions. Occasionally, BADAS presents with features overlapping with other neutrophilic dermatoses, such as Sweet syndrome or pyoderma gangrenosum. Gastrointestinal symptoms are often present, especially in IBD-related cases. Histopathology is clear but not specific respective to the other neutrophilic dermatoses. The diagnosis of BADAS is challenging, but its relapsing–remitting course and the strong association with polyarthralgia and gastrointestinal disease may assist in the diagnosis. Clinicians should be alert to cutaneous lesions manifesting together with articular symptoms in the context of gastrointestinal conditions.

## Figures and Tables

**Figure 1 metabolites-13-00790-f001:**
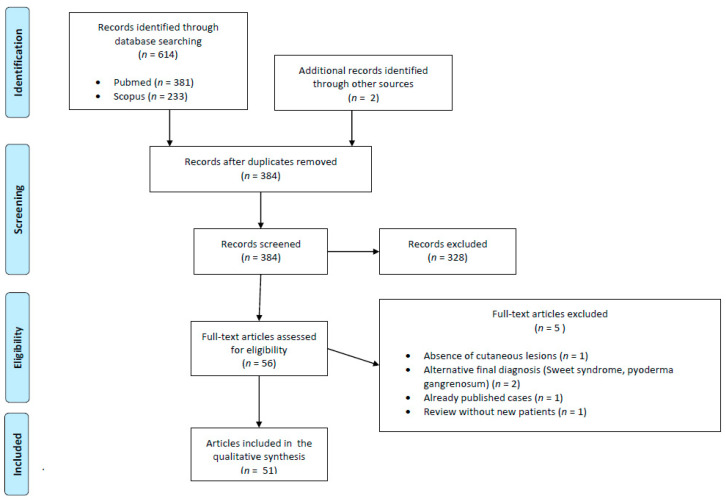
PRISMA flow diagram detailing the literature search and study selection process for systematic review.

**Figure 2 metabolites-13-00790-f002:**
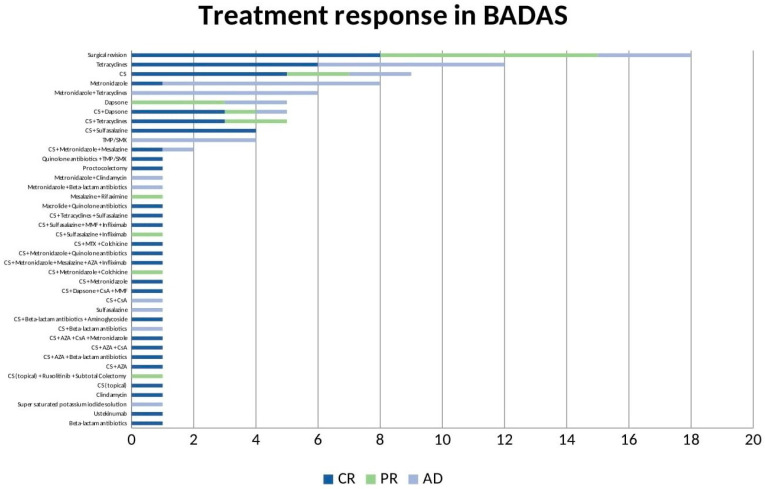
Treatment response in BADAS. CR = complete remission; PR = partial remission; AD = active disease. CS = corticosteroid; TMP/SMX = trimethoprim/sulfamethoxazole; CsA = cyclosporine A; MMF = mycophenolate mofetil; MTX = methotrexate; AZA = azathioprine.

**Table 1 metabolites-13-00790-t001:** Epidemiology, associated diseases and clinical course.

		Reported Patients
**Epidemiology**		
Mean age, years (range) ± SD	39.7 (4–78) ± 14.1	77
Male (*n*, %)	16 (20.8)	77
**Associated medical gastrointestinal diseases (*n, %*)**	32 (28.3)	113
Inflammatory bowel disease (*n, %*)	28 (24.7)	113
Crohn’s disease (*n, %*)	11 (39.3)	28 *
Ulcerative colitis (*n, %*)	16 (57.1)	28 *
Others (1 small intestinal bacterial overgrowth, 1 appendicitis, 1 diverticulitis, 1 cystic fibrosis)	4 (3.5)	113
**Post-surgery (*n, %*)**	81 (71.6)	113
Obesity (*n, %*)	72 (63.7)	113
Gastroduodenal peptic ulcers (*n, %*)	7 (6.2)	113
Others (traumatic, congenital aganglionic megacolon) (*n, %*)	2 (1.8)	113
**IBD diagnosed at the moment of BADAS presentation (*n, %*)**	9 (32.1)	28
**Latency from gastrointestinal disease onset/surgery when diagnosis of BADAS is subsequent**		
≤1 year (*n, %*)	28 (46.7)	60
1–5 y (*n, %*)	17 (28.3)	60
≥5 y (*n, %*)	15 (25)	60
**History of previous relapsing–remitting cutaneous–articular episodes (*n, %*)**	36 (52.1)	69

* One patient with indeterminate IBD.

**Table 2 metabolites-13-00790-t002:** Clinical features.

	IBD (28)	Non IBD (85)	
**Typical presentation (urticarial maculopapules with vesicopustules, erosions, healing without scars) (*n, %*)**	96 (85)		113
	26	70	
**Other presentations**			
Ulcerated noduloplaques on dorsal hands (*n, %*)	2 (1.8)		113
	1	1	
Urticarial lesions without vesicopustules (*n, %*)	6 (5.3)		113
	0	6	
Purpuric base (*n, %*)	8 (7.0)		113
	6	2	
Cellulitis/fasciitis-like on lower limbs (*n, %*)	2 (1.8)		113
	1	1	
Pyoderma-gangrenosum-like lesions (*n, %*)	2 (1.8)		113
	2	0	
Lip ulcerations (*n, %*)	3 (2.6)		113
	2	1	
Erythema-nodosum-like lesions (*n*, %)	16 (14.2)		113
Erythema-nodosum-like presentation without other lesions	9 (8)		113
**Fever (*n, %*)**	54 (49.5)		109
**Polyarthralgia (*n, %*)**	52 (46)		113
**Localized arthritis (*n, %*)**	76 (64.4)		113
**Sites of objective arthritis ***			76
Hand (*n, %*)	33 (43.4)		76
Knee (*n, %*)	24 (31.6)		76
Ankle (*n, %*)	18 (23.7)		76
Elbow (*n, %*)	17 (22.4)		76
**Gastrointestinal symptoms associated with cutaneous disease (*n, %*)**	39 (34.5)		113
	19	20	
**Cutaneous localization of typical lesions (*n*, %)**			
Upper extremities	61 (73.5)		83
Lower extremities	56 (67.5)		83
Trunk	44 (53)		83

* Other sites: shoulder 21.1% (16/76), wrist 18.4% (14/76), hip 11.8 (9/76), foot 5.3% (4/76), sacroiliac joint 4.0% (3/76).

**Table 3 metabolites-13-00790-t003:** Laboratory features.

		Reported Patients
**Histological features**		
Neutrophilic dermal and/or epidermal infiltrate (*n*, %)	56 (96.6)	58
Papillary dermis edema (*n, %*)	20 (34.5)	58
Presence of eosinophils (*n*, %)	9 (15.5)	58
Signs of vasculitis, e.g., fibrinoid necrosis (*n*, %)	13 (22.4)	58
Neutrophil adnexotropism (*n, %*)	5 (8.6)	58
**Direct immunofluorescence positivity (*n*, %)**	23 (65.7)	35
Along basement membrane zone (*n*, %)	18 (51.4)	35
Along dermal vessels (*n*, %)	11 (31.4)	35
C3 deposits (*n*, %)	17 (48.6)	35
IgG deposits (*n*, %)	12 (34.3)	35
IgM deposits (*n*, %)	12 (34.3)	35
**Laboratory blood tests**		57
Leukocytosis and/or neutrophilia (*n*, %)	23 (40.3)	57
Raised *velocity* of erythrocyte sedimentation (*n*, %)	38 (66.6)	57
Raised C-reactive protein (*n*, %)	28 (49.1)	57
Anemia (*n*, %)	21 (36.8)	57

## Data Availability

Anonymized data will be shared upon reasonable request from any qualified investigator for purposes of replicating procedures and results due to privacy.

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
