# Peer review of "The Bowel-Associated Arthritis–Dermatosis Syndrome (BADAS): A Systematic Review"

_metabolites, 2023, doi:10.3390/metabo13070790_

Round 1

Reviewer 1 Report

Extensive cover of literature and profound analysis are the advantages of the review. Bariatric surgery is increasingly used due to the ubiquity of metabolic syndrome. The number of patients with inflammatory bowel diseases also continues to increase annually. This is the main reason why the work is so important. The data are given in tables. The review summarizes all research in the area. It can be recommended for print.

Author Response

Thank you very much for the comments, we are happy that the work has been appreciated.

Reviewer 2 Report

This is a nice systematic review on an uncommon dermatosis potentially affecting patients with gastrointestinal disease. Since this condition is quite unusual, the review may be helpful to remind clinicians to consider BADAS in the differential, particularly when caring for patients with IBD, whose extraintestinal skin manifestations may have pleiotropic appearance.

The review is well-written and the methodology is sound. 

I'm only wondering whether the Authors may add some pictures from their own experience to further enhance the manuscript.

Well-written. There are only a few typos and minor errors.

Author Response

Thank you for your comments. We added a picture of our own experience.

Reviewer 3 Report

The authors summarize and critically evaluate the existing literature on BADAS and provide a comprehensive overview of the current understanding of this disorder, including its epidemiology, clinical manifestations, histopathology and management strategies.  There are a few ways that the authors can improve the manuscript.

1. What is a “qualitative synthesis” in terms of its definition, methodology, and need for PRISMA guidelines?

2. How was “methodological quality” judged in this qualitative synthesis, e.g., according to what criteria, and how was it used in analyses or discussion?  

3. How was “risk of bias” judged in this qualitative synthesis, e.g., according to what criteria, and how was it used in analyses or discussion?  

4. What biases could possibly distort your data or your conclusions?  Publication bias?  What else?

5. How was missing / incomplete data dealt with in analyses?  What were your options regarding this? 

6.  In section 2.4, please consider citing the reports that are being discussed.

7. Some minor typos, for example, top of Table 1, is 39,7 supposed to be 39.7?

8. In the “epidemiology” section of the Discussion and elsewhere in the manuscript, the authors use the word “causes” when referring to, for example, IBD.  I think this is OK, but given the authors state the pathogenesis is not fully understood, I would recommend the authors be completely comfortable that the association is clearly and indisputably causal (as opposed to coincidental related to another underlying cause, or part of an undefined causal network).  Also, has this condition been increasing or decreasing over time?  Any other relevant epidemiological features would be helpful here. 

9. The rationale for the manuscript, other than there has not been a review of BADAS, could be clearer.  What specifically do the authors hope to achieve through this article?  Then this could be highlighted in the conclusion as well – what should clinicians, researchers, administrators, etc., do with this information?  

Some minor typos, for example, top of Table 1, is 39,7 supposed to be 39.7?

Author Response

Thank you for your suggestions.

  • We have expanded the two paragraphs “Data extraction and analysis” and “Quality, limitations and Risk of Bias”. We have provided a clearer description of the selection methodology and specified how the data was synthesized. We have highlighted additional biases and included references to articles we discussed as limitations.
  • We used the term 'cause' for IBD and bariatric surgery, while for other diseases reported only anecdotally, we used 'associated.' Additionally, we described how the epidemiology of IBD and consequently of BADAS has changed.
  • We highlighted the rationale of the manuscript in the conclusion.
  • We corrected the minor typos that you noticed.

Reviewer 4 Report

The paper presents a systematic review of problem, which is rarely discussed in the literature, namely the Bowel-Associated Arthritis-Dermatosis Syndrome (BADAS). The work is interesting and methodologically correct. However, a few points need clarification:

  • What is the association between topic of the manuscript and the scope of the journal „Metabolites”?
  • Please add some keywords
  • Results are presented as tables only. Commentary on findings for a given feature should be included as text.
  • Is there a correlation between the type of histopathological changes and the type of skin lesions?
  • Ruxolitinib is not a biological drug
  • There is no comment about the limitations of the work, in particular based on case reports and reporting of the analyzed feature, e.g. the result of histopathology, only in some cases, sometimes constituting only half of the analyzed group

Author Response

Dear reviewer, here our replies.

  • The journal Metabolites has a special issue “Skin Metabolism and Cutaneous Disorders”,which we deemed relevant to our article.
  • We have added some keywords.
  • We have added the comments in the Result section.
  • We have expanded the Limitations section.
  • We have corrected the definition of Ruxolitinib.
  • The correlations between the type of histopathological changes and the type of skin lesions are reported in section 4.4.